# Success at any cost: value constrained model-free continuous control

## Abstract

Naively applying Reinforcement Learning algorithms to continuous control problems – such as locomotion and robot control – to maximize task reward often results in policies which rely on high-amplitude, high-frequency control signals, known colloquially as *bang-bang* control. While such policies can implement the optimal solution, particularly in simulated systems, they are often not desirable for real world systems since bang-bang control can lead to increased wear and tear and energy consumption and tends to excite undesired second-order dynamics. To counteract this issue, multi-objective optimization can be used to simultaneously optimize both the reward and some auxiliary cost that discourages undesired (e.g. high-amplitude) control. In principle, such an approach can yield the sought after, smooth, control policies. It can, however, be hard to find the correct trade-off between cost and return that results in the desired behavior. In this paper we propose a new constraint-based approach which defines a lower bound on the return while minimizing one or more costs (such as control effort). We employ Lagrangian relaxation to learn both (a) the parameters of a control policy that satisfies the desired constraints and (b) the Lagrangian multipliers for the optimization. Moreover, we demonstrate policy optimization which satisfies constraints either in expectation or in a per-step fashion, and we learn a single conditional policy that is able to dynamically change the trade-off between return and cost. We demonstrate the efficiency of our approach using a number of continuous control benchmark tasks as well as a realistic, energy-optimized quadruped locomotion task.[1]

## 1 Introduction

Deep Reinforcement Learning (RL) has achieved great successes over the last couple of years, enabling learning of effective policies from high-dimensional input, such as pixels, on complicated tasks. However, compared to problems with discrete action spaces, continuous control problems with high-dimensional continuous state-action spaces – as often encountered in robotics – have proven much more challenging. Beyond the issue of exploration in high-dimensional continuous action spaces, RL algorithms rarely learn policies that produce smooth control signals when just optimizing for success. Instead, policies often exhibit control signals that switch between extreme values at high-frequency, often colloquially referred to as *bang-bang* control. Smoothness, however, is a desirable property in most real-world control problems. Unnecessary oscillations are not only energy inefficient, they also exert stress on a physical system by exciting second-order dynamics and increasing wear and tear on structural elements and actuators.

To regularize the behavior, one can add penalties to the reward function. As a result, the reward function is composed of positive reward for achieving the goal and negative reward (penalties) for control action discontinuities or high energy use. This effectively casts the problem into a multi-objective optimization setting, where – depending on the ratio between the reward and the different penalties – different behaviors may be achieved. While every ratio will have its optimal policy, finding the ratio that results in the desired behavior, i.e. smooth control while still achieving an acceptable task success rate, is not always trivial and requires excessive hyperparameter tuning. Often, one must find different hyperparameter settings for different reward-penalty trade-offs or tasks. The process of finding these parameters is tedious and cumbersome, and may prevent robust

---

[1]Videos available at `https://sites.google.com/view/minitauriclr2019`

general solutions. In this paper we rephrase the problem: instead of trying to find the right ratios between reward and penalties, we regularize the optimization problem by adding constraints, thereby reducing its effective dimensionality. More specifically, we propose to minimize the penalty with respect to a lower bound on the success rate of the task.

Using a Lagrangian relaxation technique, we introduce cost coefficients for each of the imposed constraints that are tuned automatically during the optimization process. In this way we can find the optimal trade-off between reward and costs (that also satisfies the imposed constraints) automatically. By making the cost multipliers state-dependent, and adapting them alongside the policy, we can not only impose constraints on expected reward or cost, but also on their instantaneous values. Such point-wise constraints allow for much tighter control over the behavior of the policy, since a constraint that is satisfied only in overall expectation could still be violated momentarily. Finally, the entire constrained optimization procedure that we introduce can furthermore be conditioned on the constraint bounds themselves in order to learn a single, bound-conditioned policy that is able dynamically trade-off reward and penalties. This allows us to, for example, learn energy-efficient locomotion at a range of different velocities.

The contributions of this work are (i) we regress state-dependent Lagrangian multipliers with a neural network in order to generalize across states, (ii) we impose structure to the critic by simultaneously learning both reward and value estimates as well as the coefficient to trade them off in a single model, and finally (iii) we train a bound-conditioned policy that is optimized for a range of bounds. Our approach, as described in more detail in Section 3, is general and flexible in that it can be applied to any value-based RL algorithm and any number of constraints. We evaluate our approach on continuous control problems in Section 4 using tasks from the DM Control Suite (Tassa et al., 2018) and a (precisely simulated) locomotion task with the Minitaur quadruped.

## 2 BACKGROUND AND RELATED WORK

We consider the classical Markov Decision Process (MDP) setting (Sutton & Barto, 1998), where an agent sequentially interacts with an environment. More precisely, the agent observes the state of the environment $s$ and decides on which action to take according to a policy $a \sim \pi(s \mid s)$. Executing the action in the environment, then, causes a state transition. Each transition has an associated reward defined by some utility function $r(s, a)$. The goal of the agent is to maximize the expected sum of rewards, also known as the return, $\max_\pi \mathbb{E}_{s,a \sim \pi} [\sum_t r(s_t, a_t)]$. While some tasks have a well-defined reward, such as the increase in score when playing a game, for many others the objective is not as easily defined. Designing reward functions that produce a desired behavior policy can thus be extremely difficult, even in the single-objective case (e.g. Popov et al., 2017; Amodei et al., 2016).

Multi-Objective RL (MORL) problems arise in many domains, including robotics, and have been covered by a rich body of literature (see e.g. Roijers et al., 2013, for a recent review), suggesting a variety of solution strategies. For instance, Mossalam et al. (2016) devise a Deep RL algorithm that implements an outer loop method and repeatedly calls a single-objective solver. Mannor & Shimkin (2004) propose an algorithm for learning in a stochastic game setting with vector valued rewards (their approach is based on approachability of a target set in the reward space). However, most of these approaches explicitly recast the multi-objective problem into a single-objective problem (that is amenable to existing methods), where one aims to find the trade-off between the different objectives that yields the desired result. In contrast, we aim for a method that automatically trades off different components in the objective to achieve a particular goal. To achieve this, we cast the problem in the framework of Constrained Markov Decision Processes (CMDPs) (Altman, 1999). CMDPs have been considered in a variety of works, including in the robotics and control literature. For instance, Achiam et al. (2017) and Dalal et al. (2018) focus on constraints motivated by safety concerns and propose algorithms that ensure that constraints remain satisfied at all times. These works, however, assume that the initial policy already satisfies the constraint, which is not the case when the constraint involves the task success rate; as in this work. The motivation for the work by Tessler et al. (2018) is similar to ours. In contrast to our work, their approach maximizes reward subject to a constraint on the cost and enforces constraints only in expectation. Additionally, as an advance over the existing literature, we explicitly learn separate values for the reward and cost, as well as state-dependent coefficients that enable us to trade off the two in the policy optimization.

Constraint-based formulations are also used frequently in single-objective policy search algorithms where bounds on the policy divergence are employed to control the rate of change in the policy from one iteration to the next (e.g. Peters & Mülling, 2010; Levine & Koltun, 2013; Schulman et al., 2015; Abdolmaleki et al., 2018). Our use of constraints, while similar in the employed methods, can be seen as orthogonal to the idea of using constraints to bound the rate of change in a policy. While we note that our approach can be applied to any value-based off-policy method, we make use of the method described in Maximum a Posteriori Policy Optimisation (MPO) (Abdolmaleki et al., 2018) as the underlying policy optimization algorithm – without loss of any generality of our method. MPO is an actor-critic algorithm that is known to yield robust policy improvement. In each policy improvement step, for each state sampled from replay buffer, MPO creates a population of actions. Subsequently, these actions are re-weighted based on their estimated values such that better actions will have higher weights. Finally, MPO uses a supervised learning step to fit a new policy in continuous state and action space. See Abdolmaleki et al. (2018) and Appendix A for more details.

## 3 CONSTRAINED OPTIMIZATION FOR CONTROL

We consider MDPs where we have both a reward and cost, $r\left(\boldsymbol{s}, \boldsymbol{a}\right)$ and $c\left(\boldsymbol{s}, \boldsymbol{a}\right)$, which are functions of state $\boldsymbol{s}$ and action $\boldsymbol{a}$. The goal is to automatically find a probabilistic policy $\pi(\boldsymbol{a}|\boldsymbol{s}; \theta)$ (with parameter $\theta$) that trades-off between maximizing the (expected) reward and minimizing the cost – in order to achieve the desired behavior. In the case of continuous control, desirable behavior would be solving the task (e.g. stable swing up in cart-pole) while minimizing other quantities, such as control effort or energy. In effect we want to optimize the total return subject to a penalty proportional to the total cost, i.e. $\max_\pi \mathbb{E}_{\mathbf{s}, \mathbf{a} \sim \pi} \left[\sum_t r\left(\boldsymbol{s}_t, \boldsymbol{a}_t\right) - \alpha \cdot c\left(\boldsymbol{s}_t, \boldsymbol{a}_t\right)\right]$, where we take $\max_\pi$ to mean maximizing the objective with respect to the policy parameters $\theta$. The expectation over states $\boldsymbol{s}$ is with respect to the state visitation probability under the policy $p^\pi\left(\boldsymbol{s}\right)$. The problem of finding the right trade-off then becomes a matter of finding a good value for $\alpha$. Finding this trade-off is often non-trivial. An alternative way of looking at this dilemma is to take a multi-objective optimization perspective. Instead of fixing $\alpha$, we can optimize for it simultaneously and can obtain different Pareto-optimal solutions for different values of $\alpha$. In addition, to ease the definition of a desirable regime for $\alpha$, one can consider imposing hard constraints on the cost to reduce dimensionality (Deb, 2014), instead of linearly combining the different objectives. Defining such hard constraints is often more intuitive than trying to manually tune coefficients. For example, in locomotion, it is easier to define desired behavior in terms of a lower bound on speed or an upper bound on an energy cost.

### 3.1 CONSTRAINED MDPs

The constrained perspective outlined above can be formalized as CMDPs (Altman, 1999). While a constraint can be placed on either the reward or the cost, in this work we consider a lower bound on the expected total return (although the theory derived below equivalently applies to constraints on cost), i.e. $\min_\pi \mathbb{E}_{\mathbf{s}, \mathbf{a} \sim \pi} \left[\sum_t c\left(\boldsymbol{s}_t, \boldsymbol{a}_t\right)\right]$, s.t. $\mathbb{E}_{\mathbf{s}, \mathbf{a} \sim \pi} \left[\sum_t r\left(\boldsymbol{s}_t, \boldsymbol{a}_t\right)\right] \geq \bar{R}$, where $\bar{R}$ is the minimum desired return. In the case of an infinite horizon with a given stationary state distribution, the constraint can instead be formulated for the per-step reward, i.e. $\mathbb{E}_{\mathbf{s}, \mathbf{a} \sim \pi} \left[r\left(\boldsymbol{s}, \boldsymbol{a}\right)\right] \geq \bar{r}$. In practice one often optimizes the $\gamma$-discounted return in both cases. To apply model-free RL methods to this problem we first define an estimate of the expected discounted return for a given policy as the action-value function $Q_r\left(\boldsymbol{s}, \boldsymbol{a}\right) = \mathbb{E}_{\mathbf{s}, \mathbf{a} \sim \pi} \left[\sum_t \gamma^t \cdot r\left(\boldsymbol{s}_t, \boldsymbol{a}_t\right) | \boldsymbol{s}_0 = \boldsymbol{s}, \boldsymbol{a}_0 = \boldsymbol{a}\right]$. Further, let $Q_c\left(\boldsymbol{s}, \boldsymbol{a}\right)$ denote the similarly constructed expected discounted cost action-value function. Equipped with these value functions, we can then recast the CMDP in value-space, where $\bar{V}_r = \bar{r} / \left(1 - \gamma\right)$ (i.e. scaling the desired reward $\bar{r}$ with the limit of the converging sum over discounts):

$$\min_\pi \mathbb{E}_{\mathbf{s}, \mathbf{a} \sim \pi} \left[Q_c\left(\boldsymbol{s}, \boldsymbol{a}\right)\right], \text{ s.t. } \mathbb{E}_{\mathbf{s}, \mathbf{a} \sim \pi} \left[Q_r\left(\boldsymbol{s}, \boldsymbol{a}\right)\right] \geq \bar{V}_r. \tag{1}$$

### 3.2 LAGRANGIAN RELAXATION

We formulate task success via a constraint on the reward. Fulfilling this constraint indicates task success. Generally the constraint is not satisfied at the start of learning, as the agent first needs to learn how to solve the task. This limits the choice of existing methods that can be used to solve the CMDP, as many of these methods assume that the constraint is satisfied at the start and limit

the policy update to remain within the constraint-satisfying regime (e.g. Achiam et al., 2017). Lagrangian relaxation is a general method for solving general constrained optimization problems; and CMDPs by extension (Altman, 1999). In this setting, the hard constraint is relaxed into a soft constraint, where any constraint violation acts as a penalty for the optimization. Applying Lagrangian relaxation to Equation 1 results in the unconstrained dual problem

$$\max_{\pi} \min_{\lambda \geq 0} \mathbb{E}_{\mathbf{s},\mathbf{a}\sim\pi} \left[Q_\lambda\left(\boldsymbol{s},\boldsymbol{a}\right)\right], \text{ with } Q_\lambda\left(\boldsymbol{s},\boldsymbol{a}\right) = \lambda\left(Q_r\left(\boldsymbol{s},\boldsymbol{a}\right) - \bar{V}_r\right) - Q_c\left(\boldsymbol{s},\boldsymbol{a}\right), \qquad (2)$$

with an additional minimization objective over the Lagrangian multiplier $\lambda$.

A larger $\lambda$ results in a higher penalty for violating the constraint. Hence, we can iteratively update $\lambda$ by gradient descent on $Q_\lambda\left(\boldsymbol{s},\boldsymbol{a}\right)$, alternated with policy optimization, until the constraint is satisfied. Under assumptions described in Tessler et al. (2018), this approach converges to a saddle point. At convergence, when $\nabla_\lambda \mathbb{E}\left[Q_\lambda\left(\boldsymbol{s},\boldsymbol{a}\right)\right] = 0$, $\lambda$ is exactly the desired trade-off between reward and cost we aimed to find. To perform the policy optimization for $\pi$ any off-the-shelf off-policy optimization algorithm can be used (since we assume that we have a learned, approximate Q-function at our disposal). In practice, we perform policy optimization using the MPO algorithm (Abdolmaleki et al., 2018) and refer to Appendix A for additional details.

**Scale invariance**  At the start of learning, as the constraint is not yet satisfied, $\lambda$ will grow in order to suppress the cost $Q_c\left(\boldsymbol{s},\boldsymbol{a}\right)$ and focus the optimization on maximizing $Q_r\left(\boldsymbol{s},\boldsymbol{a}\right)$. Depending on how quickly the constraint can be satisfied, $\lambda$ can grow very large, resulting in a overall large magnitude of $Q_\lambda\left(\boldsymbol{s},\boldsymbol{a}\right)$. This can result in unstable learning as most actor-critic methods that have an explicit parameterization of $\pi$ are especially sensitive to large (swings in) values. To improve stability, we re-parameterize $Q_\lambda\left(\boldsymbol{s},\boldsymbol{a}\right)$ to be a projection into a convex combination of $(Q_r\left(\boldsymbol{s},\boldsymbol{a}\right) - V_r^*)$ and $-Q_c\left(\boldsymbol{s},\boldsymbol{a}\right)$. Instead of scaling only the reward term, we can then adaptively reweigh the relative importance of reward and cost, and make the magnitude of $Q_\lambda\left(\boldsymbol{s},\boldsymbol{a}\right)$ bounded. To enforce $\lambda \geq 0$, we can perform a change of variable $\lambda' = \log\left(\lambda\right)$ to obtain the following dual optimization problem

$$\max_{\pi} \min_{\lambda' \in \mathbb{R}} \mathbb{E}_{\mathbf{s},\mathbf{a}\sim\pi} \left[Q_{\lambda'}\left(\boldsymbol{s},\boldsymbol{a}\right)\right], \text{ with } Q_{\lambda'}\left(\boldsymbol{s},\boldsymbol{a}\right) = \frac{\exp\left(\lambda'\right)\left(Q_r\left(\boldsymbol{s},\boldsymbol{a}\right) - V_r^*\right) - Q_c\left(\boldsymbol{s},\boldsymbol{a}\right)}{\exp\left(\lambda'\right) + 1}. \qquad (3)$$

Note that to correspond to the formulation in Equation 2, we only perform gradient descent w.r.t. $\lambda'$ on the first term in the numerator. In practice, we limit $\lambda'$ to $[\lambda'_{\min}, \lambda'_{\max}]$, with $\left(\exp\left(\lambda'_{\max}\right) + 1\right)^{-1} = \epsilon$ for some small $\epsilon$, and initialize to $\lambda'_{\max}$.

### 3.3  POINT-WISE CONSTRAINTS

One downside of the CMDP formulation given in Equation 1 is that the constraint is placed on the *expected* total episode return, or *expected* reward. This implies that the constraint will not necessarily be satisfied at every single timestep, or visited state, during the episode. For some tasks this difference, however, turns out to be of importance. For example, in locomotion, a constant speed is more desirable than a fluctuating one, even though the latter might also satisfy a minimum velocity in expectation. Fortunately, we can extend the single constraint introduced in Section 3.1 to a set, possibly infinite, of point-wise constraints; one for each state induced by the policy. This can be formulated as the following optimization problem:

$$\min_{\pi} \mathbb{E}_{\mathbf{s},\mathbf{a}\sim\pi} \left[Q_c\left(\boldsymbol{s},\boldsymbol{a}\right)\right], \text{ s.t. } \forall \mathbf{s} \sim \pi : \mathbb{E}_{\mathbf{a}\sim\pi}\left[Q_r\left(\boldsymbol{s},\boldsymbol{a}\right)\right] \geq \bar{V}_r. \qquad (4)$$

Analogous to Section 3.2, this problem can be optimized with Lagrangian relaxation by introducing state-dependent Lagrangian multipliers. Formally, we can write,

$$\max_{\pi} \mathbb{E}_{\mathbf{s}\sim\pi} \left[\min_{\lambda(\boldsymbol{s})\geq 0} \mathbb{E}_{\mathbf{a}\sim\pi}\left[Q_\lambda\left(\boldsymbol{s},\boldsymbol{a}\right)\right]\right], \text{ with } Q_\lambda\left(\boldsymbol{s},\boldsymbol{a}\right) = \lambda\left(\boldsymbol{s}\right)\left(Q_r\left(\boldsymbol{s},\boldsymbol{a}\right) - \bar{V}_r\right) - Q_c\left(\boldsymbol{s},\boldsymbol{a}\right). \quad (5)$$

Analogously to how one often assumes that nearby states have a similar value, here we have made the assumption that nearby states have similar $\lambda$ multipliers. This allows learning a parametric function $\lambda\left(\boldsymbol{s}\right)$ alongside the action-value, which can generalize to unseen states $\boldsymbol{s}$. In practice, we train a single critic model that outputs $\lambda\left(\boldsymbol{s}\right)$ as well as $Q_c\left(\boldsymbol{s},\boldsymbol{a}\right)$ and $Q_r\left(\boldsymbol{s},\boldsymbol{a}\right)$. We provide pseudocode for

the resulting constrained optimization algorithm in Appendix A. Note that, in this case, the lower bound is still a fixed value and does not depend on the state. In general such a constraint might be impossible to satisfy for some states in a given task if the state distribution is not stationary (e.g. we cannot satisfy a reward constraint in the swing-up phase of the simple pendulum). However, the lower bound can also be made state-dependent and our approach will still be applicable.

### 3.4 Conditional constraints

Up to this point, we have made the assumption that we are only interested in a single, fixed value for the lower bound. However, in some tasks one would want to solve Equation 4 for different lower bounds $\bar{V}_r$, i.e. minimizing cost for various success rates. For example, in a locomotion task, one could be interested in optimizing energy for multiple different target speeds or gaits. Assuming locomotion is a stationary behavior, one could set $\bar{V}_r = \bar{v}/(1-\gamma)$ for a range of velocities $\bar{v} \in [0, v_{\max}^-]$. In the limit this would achieve the same result as multi-objective optimization–it would identify the set of solutions wherein it is impossible to increase one objective without worsening another–also known as a Pareto front. To avoid the need to solve a large number of optimization problems, i.e., solving for every $\bar{V}_r$ separately, we can condition the policy, value function and Lagrangian multipliers on the desired target value and, effectively, learn a bound-conditioned policy

$$\mathbb{E}_{z \sim p(z)} \left[ \max_{\pi(z)} \mathbb{E}_{\mathbf{s} \sim \pi(z)} \left[ \min_{\lambda(s,z) \geq 0} \mathbb{E}_{\mathbf{a} \sim \pi(z)} \left[ Q_\lambda \left( s, a, z \right) \right] \right] \right],$$
$$\text{with } Q_\lambda \left( s, a, z \right) = \lambda \left( s, z \right) \left( Q_r \left( s, a, z \right) - \bar{V}_r \left( z \right) \right) - Q_c \left( s, a, z \right). \quad (6)$$

Here $z$ is a goal variable, the desired lower bound for the reward, that is observed by the policy and critic and maps to a lower bound for the value $\bar{V}_r \left( z \right)$. Such a conditional constraint allows a single policy to dynamically trade off cost and return.

## 4 Experiments

We apply our constraint-based approach to the continuous control domains shown in Figure 1: the cart-pole and humanoid from the DM Control Suite benchmark, and a more challenging robot locomotion task.



|  (a) Cart-pole | (b) Humanoid | (c) Minitaur |

Figure 1: The continuous control environments used in the experiments. Cart-pole swingup (a) and humanoid stand and walk (b) are from the DM control suite (Tassa et al., 2018). The Minitaur robot (c) is similarly simulated in MuJoCo. The red dot denotes the IMU.

### 4.1 Control benchmarks

We consider three tasks from the DeepMind Control Suite (Tassa et al., 2018) benchmark to illustrate the problem of bang-bang control and the effectiveness of our approach: cart-pole swingup, humanoid stand and humanoid walk. Each of these tasks has a shaped reward that combines the success criterion (e.g. pole upright and cart in the center for cart-pole) with a bonus for a low control signal. The total reward lies in $[0, 1]$ in all cases. We compare agents trained on this original reward with two that are trained with the control term from the reward removed, one unconstrained that never observes a control penalty, and one constrained where we minimize the control penalty with respect to a lower bound on return using the approach detailed in the previous section. In all cases we train a neural network controller using the MPO algorithm (Abdolmaleki et al., 2018). More

Table 1: Average reward and penalty for the different control benchmark tasks and policies trained in the constrained, unconstrained and original reward setup. In all cases, the constraint-based approach results in the lowest average penalty. While the lower bound was set to $0.9$ of a maximum of $1$, we obtain the same average reward as the unconstrained case for the cartpole swingup and humanoid stand tasks.

| Task | Window | Constrained | | Unconstrained | | Original | |
|------|--------|-------------|---------|---------------|---------|----------|---------|
| | | reward | penalty | reward | penalty | reward | penalty |
| cartpole | full | 0.891 | **0.302** | 0.885 | **1.918** | 0.895 | 0.733 |
| | last 50% | 0.998 | **0.013** | 1.000 | **1.459** | 0.998 | 0.074 |
| humanoid (stand) | full | 0.961 | **5.608** | 0.964 | **37.189** | 0.952 | 27.518 |
| | last 50% | 0.998 | **4.538** | 0.993 | **37.288** | 0.999 | 27.007 |
| humanoid (walk) | full | 0.869 | **21.595** | 0.953 | 26.835 | 0.957 | **29.565** |
| | last 50% | 0.903 | **21.295** | 0.984 | 26.819 | 0.990 | **29.418** |

specifically, we train a two-layer MLP policy to output the mean and variance of a Gaussian policy. We use a fixed lower bound on the expected per-step reward of $0.9$ and use the norm of the force output as the penalty to minimize. More details about the training setup can be found in Appendix A. Table 1 shows the average reward (excl. control penalty) and control penalty for each of the tasks and setups, both averaged across the entire episode as well as the final 50%. The latter is relevant as all three tasks have an initial balancing component that by its nature requires significant control input.

For cart-pole, we see that all agents give almost identical returns but the constrained method is able to achieve significantly lower penalties, even compared to the original reward that included a (non-adaptive) penalty (over 50% across the entire episode, over 80% in the final half of the episode). Even though the lower bound on the reward is only $0.9$, the constrained method still achieves higher rewards because after the swingup phase, the best thing to do in order to minimize the control input is to keep the pole balanced. Figure 2a shows a typical execution of the noisy policy when optimizing for the reward alone. Note that actions are clamped in $[-1, 1]$. We can observe that the average absolute control signal is large and the agent keeps switching rapidly between a large negative and large positive force even after the swing up phase. While the agent is able to

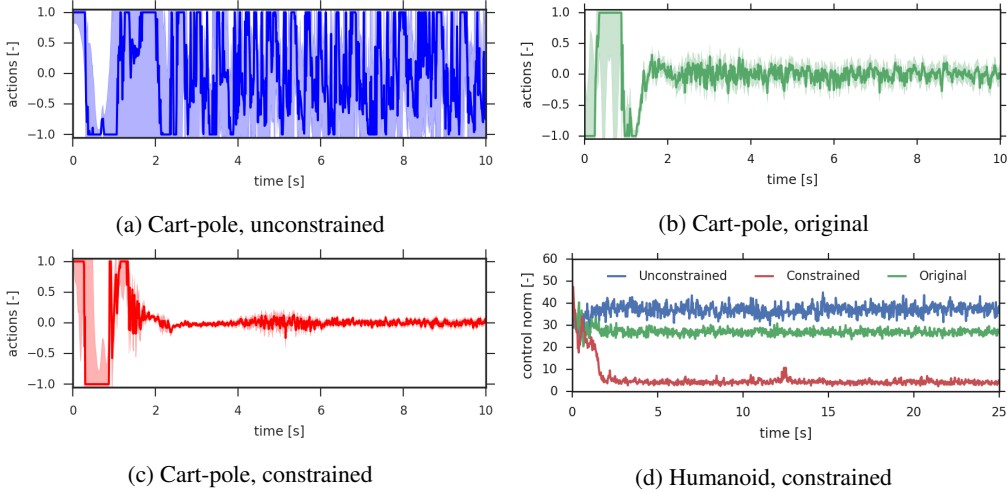

(a) Cart-pole, unconstrained

(b) Cart-pole, original

(c) Cart-pole, constrained

(d) Humanoid, constrained

Figure 2: Representative results of the executed policies in the control benchmark tasks. Plots (a), (b) and (c) show the mean and standard deviation as output by the policy trained on cart-pole swingup in the constrained, unconstrained and original reward setting respectively, following a trajectory generated using actions sampled from this distribution. In all three cases, we observe high control input during the first 2 seconds, corresponding to the swingup phase. Figure (d) shows the control norm during the episode rollout of policies trained in humanoid stand. Note that in all cases the actual return between the thee methods is almost identical.

solve the task (and the behaviour can be somewhat smoothed by executing only the mean of the learned Gaussian policy for this simple system), this kind of *bang-bang* control is not desirable for any real-world control system. Figure 2c shows a typical execution of a policy learned with the constrained approach. It is clearly visible that the policy is much smoother; in particular it never reaches maximum or minimum actuation levels after the swing up (during which a switch between maximum and minimum actuation is indeed the optimal solution). Figure 2b shows the execution of the agent trained against the original reward function that also includes a (fixed) control penalty. As in the constrained case, the action distribution shrinks after the swingup phase. However, there is still more switching present compared to the constraint-based approach.

We observe a similar trend for the humanoid stand task, where all three setups result in almost the same average reward, but the constraint-based approach is able to reduce the control penalty by 80% compared to the original reward setup. We visualize the resulting policies in Figure 3a-c by over-laying frames from the final 50% time steps of the episode. A policy exhibiting bang-bang control will result in more jittering motion and hence a more blurry image. As can be seen in Figure 3a, the unconstrained case results in a lot of jittering motions. Both the constrained and original setup show significantly less jitter, with the humanoid adopting a fixed pose. In the constrained case, however, the agent consistently learn to adopt a pose with smaller control norm by putting the legs closer together. The same observations can be made by looking at the control norm during the episode in Figure 2d. After the initial standup phase, the contrained optimization approach results in a significantly lower control norm during the remainder of the episode. For the humanoid walk task, we observe a different result: while the constraint-based approach still results in a lower penalty, there is also a reduction in the average reward. This is to be expected: when walking, the best thing to do to minimize the penalty is to slow down, which will reduce the reward. As a result the reward will stick closer to the imposed lower bound of 0.9. Note that we do effectively satisfy the lower bound after the standup phase. Interestingly, the agent trained on the original reward configuration results in a higher control penalty compared to the unconstrained case. It is worth noting that the control penalty is mixed into the reward differently than in the (un)constrained case and may hence have a different optimum.

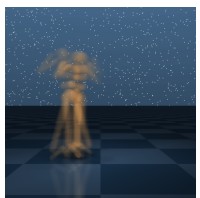 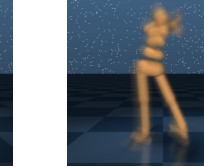 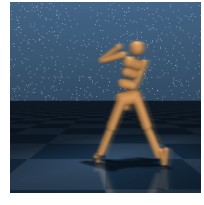

(a) Unconstrained      (b) Constrained      (c) Original

Figure 3: Comparison of policies trained on the humanoid stand task in the constrained, unconstrained and original reward setup. Figures show the average frame of the final 50% of the episode. Policies that exhibit more bang-bang-style control will result in more jittering movements and hence more blurry images. The constrained and original reward policy clearly show less jitter.

## 4.2 MINITAUR LOCOMOTION

Our second experiment is based on the the Minitaur robot developed by Ghost Robotics (Kenneally et al., 2016). It is a quadruped with two Degrees of Freedom (DoFs) in each of the four legs. High-power direct-drive actuators are used for each joint, allowing the robot to express a multitude of dynamic gaits such as trotting, pronking and galloping. These gaits, however, require a large engineering effort when implemented using state-of-the-art control techniques, and, when model-based approaches are used, performance becomes sensitive to modeling errors. Learning-based approaches have shown promise as an alternative for devising locomotion controllers for the Minitaur (Tan et al., 2018). Learning approaches are less dependent on gait and other task dependent heuristics and can lead to more versatile and very dynamic behaviors. We do however want learned gaits to be sufficiently well-behaved, avoiding high-frequency changes or large steps in the control signal that cause vibrations which ultimately can lead to control instability or mechanical stress. One way to achieve smooth control and locomotion is to optimize for energy efficiency, as fast, opposing actions typically require more power. We hence adopt an energy penalty in the following.

### 4.2.1 EXPERIMENTAL SETUP

Although the Minitaur experiments are conducted in simulation, we have made a significant effort to capture many of the challenges of real robots: physical robot complexity, realistic and partial observations, control latency, plus additional perturbations, variations, and noise. We model the Minitaur in MuJoCo (Todorov et al., 2012) as seen in Figure 1c, using model parameters obtained from data sheets as well as system identification to improve the fidelity. The Minitaur is placed on a varying, rough terrain that is procedurally generated for every rollout. To model the drive train we use a non-linear actuator model based on a general DC motor model and the torque-current characteristic described in De & Koditschek (2015). The observations of the RL agent include noisy motor positions, yaw, pitch, roll, and angular velocities and accelerometer readings, but no direct perception of the surroundings or terrain. The policy outputs position setpoints at 100Hz that are fed to a low-level proportional position controller running at 1KHz, with a forced delay of 20ms added between sensor readings and the corresponding control signal, to match delays observed on the real hardware. To improve control robustness and with the aim to transfer the controllers from simulation to real hardware, we perform domain randomization (Tobin et al., 2017) on a number of model parameters, as well as apply random external forces to the body (see Appendix B for details).

As we are only considering forward locomotion, we set the reward $r\left(s, a\right)$ to be the forward velocity of the robot's base expressed in the world frame. The cost $c\left(s, a\right)$ is set to be the total power usage of the motors according to the actuator model. As the legs can collide with the main body when giving the agent access to the full control range, a constant penalty is added to the penalty computed from the power consumption during any self-collision. We use a largely similar training setup as in Section 4.1; however, since the episodes are 30sec in length and only partial and noisy observations are available, the agent requires memory for effective state estimation, so we add an LSTM to the model. In addition to learning separate values for $Q_r\left(s, a\right)$ and $Q_c\left(s, a\right)$, we split up $Q_c\left(s, a\right)$ into separate value functions for the power usage and collision penalty. We also increase the number of actors to 100 to sample a larger number of domain variations more quickly. More details can be found in Appendix A.

### 4.2.2 RESULTS

We first look at the effect of applying the lower bound to each individual state instead of on the global average velocity. Figure 4 shows a comparison between learning dynamics between a model using a single $\lambda$ multiplier and a model with a state-dependent one, i.e. constraint in expectation or per-step. Both agents try to achieve a lower bound on the value that is equivalent to a minimum velocity of 0.5m/s. At first, both agents "focus" on satisfying the constraint, increasing the penalty significantly in order to do so. Once the target velocity is exceeded, the agents start to optimize the penalty, which drives them back to the imposed bound. A single multiplier that is applied to all

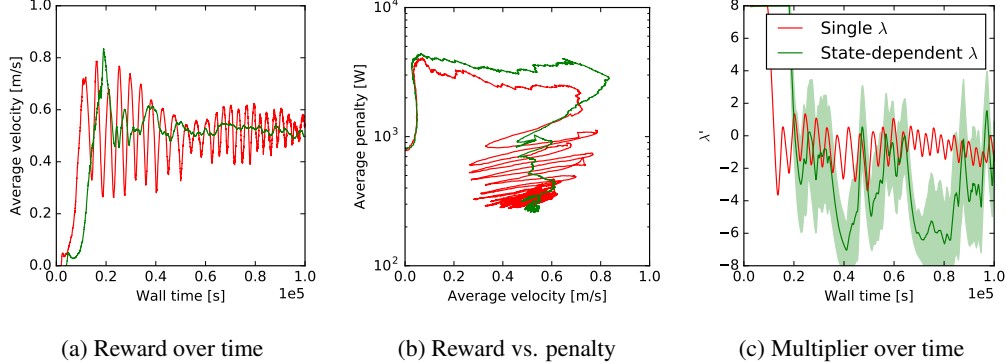

(a) Reward over time      (b) Reward vs. penalty      (c) Multiplier over time

Figure 4: Comparison of a single versus a state-dependent $\lambda$ multiplier for models trained to achieve a minimum velocity of 0.5m/s. A single multiplier results in large swings in reward and on average higher values of $\lambda$. In b, policies start off at 0m/s and first learn to satisfy the constraint before optimizing the penalty. In c, for the state-dependent case, we show the mean and standard deviation of $\lambda$ across the training batch.

Table 2: Results for models trained to achieve a fixed lower bound on the velocity. Reported numbers are average per-step (velocity overshoot [m/s], penalty [W]), except for the unbounded case where we report actual velocity. Each entry is an average over 4 seeds. We highlight the best constant $\alpha$, in terms of smallest overshoot, for each target bound.

| Target | $\alpha = 3\mathrm{e}{-3}$ | | $\alpha = 1\mathrm{e}{-3}$ | | $\alpha = 3\mathrm{e}{-4}$ | | $\alpha = 1\mathrm{e}{-4}$ | | Constraint | |
|--------|-------|---------|-------|---------|-------|---------|-------|---------|-------|---------|
| | delta | penalty | delta | penalty | delta | penalty | delta | penalty | delta | penalty |
| 0.1 | -0.1, | 35.74 | **-0.01**, | **104.2** | 0.07, | 112.35 | 0.1, | 245.49 | **0.01**, | **127.14** |
| 0.2 | -0.2, | 46.48 | **-0.01**, | **210.04** | 0.15, | 207.19 | 0.23, | 399.83 | **0.03**, | **106.88** |
| 0.3 | -0.3, | 50.3 | **0.06**, | **154.91** | 0.16, | 213.1 | 0.24, | 429.6 | **0.04**, | **89.97** |
| 0.4 | -0.4, | 54.05 | **0.06**, | **195.98** | 0.11, | 306.1 | 0.32, | 627.66 | **0.05**, | **132.97** |
| 0.5 | -0.5, | 60.71 | **0.13**, | **250.69** | **0.13**, | **332.53** | 0.26, | 808.38 | **0.05**, | **142.93** |
| $\infty$ | *0.0*, | 54.63 | *1.25*, | 775.08 | *1.24*, | 1556.97 | *1.24*, | 1656.42 | -, | - |

Table 3: Results of models that are conditioned on the target velocity, evaluated for for different values. Reported numbers are average per-step (velocity overshoot [m/s], penalty [W]). Each row is an average over 4 seeds. The highlighted numbers mark the best individual alpha for each target velocity (in terms of velocity overshoot).

| Target | $\alpha = 3\mathrm{e}{-3}$ | | $\alpha = 1\mathrm{e}{-3}$ | | $\alpha = 3\mathrm{e}{-4}$ | | $\alpha = 1\mathrm{e}{-4}$ | | Constraint | |
|--------|-------|---------|-------|---------|-------|---------|-------|---------|-------|---------|
| | delta | penalty | delta | penalty | delta | penalty | delta | penalty | delta | penalty |
| 0.0 | **0.0**, | **53.68** | 0.01, | 116.59 | 0.17, | 272.45 | 0.37, | 757.53 | **0.0**, | **84.07** |
| 0.1 | -0.1, | 54.49 | **0.0**, | **158.68** | 0.21, | 324.16 | 0.37, | 619.3 | **0.0**, | **141.86** |
| 0.2 | -0.2, | 53.54 | **0.02**, | **256.68** | 0.21, | 373.13 | 0.36, | 627.19 | **0.04**, | **174.79** |
| 0.3 | -0.3, | 53.6 | **-0.02**, | **314.71** | 0.16, | 336.48 | 0.42, | 747.24 | **0.02**, | **188.18** |
| 0.4 | -0.4, | 54.82 | **-0.07**, | **384.94** | 0.15, | 467.21 | 0.32, | 870.34 | **0.05**, | **252.54** |
| 0.5 | -0.5, | 52.37 | -0.1, | 366.48 | **0.01**, | **594.36** | 0.27, | 1026.3 | **0.05**, | **361.16** |
| 0.6 | -0.6, | 52.36 | -0.2, | 686.36 | -0.07, | 770.67 | **0.02**, | **1632.96** | -0.04, | **773.79** |

states leads to larger changes in behavior space, where the agent oscillates between moving too slow at a lower penalty or too fast at a higher penalty. The agent with the state-dependent multiplier tracks the target velocity more closely, and achieves slightly lower penalties. Looking at the $\lambda$ values over time in Figure 4c, we see that they are generally lower in the latter case as well.

In Table 2, we compare the reward-penalty trade-off for settings trained to achieve a fixed lower bound on the velocity. We compare our approach to baselines where we clip the reward as $r'\left(\boldsymbol{s}_t, \boldsymbol{a}_t\right) = \min\left(r\left(\boldsymbol{s}_t, \boldsymbol{a}_t\right), \bar{r}\right)$ and use a fixed coefficient $\alpha$ for the penalty. As there is less incentive for the agent to increase the reward over $\bar{r}$, there is more opportunity to optimize the penalty. Results shown are the per-step overshoot with respect to the desired target velocity and the penalty, averaged across 4 seeds and 100 episodes each (the first 100ms is clipped to disregard transient behavior when starting from a stand-still). We also compare to a baseline where the reward is unbounded, marked as $\infty$ in Table 2. In the unbounded reward case, we observe that it is difficult to achieve a positive but moderately slow speed. Either $\alpha$ is too high and the agent is biased towards standing still, or it is too low and the agent reaches the end of the course before the time limit (corresponding to an average velocity of approx. 1.25m/s). For the clipped reward, we observe a similar issue when $\alpha$ is set too high. In nearly all other cases, the targeted speed is exceeded by some margin that increases with decreasing $\alpha$. While there is less incentive to exceed $\bar{r}$, a larger margin decreases the chances of the actual speed momentarily dropping below the target speed. Using the constraint-based approach, we generally achieve average actual speeds closer to the target speed and at a lower average penalty, showing the merits of adaptively trading of reward and cost.

Table 3 shows a comparison between agents are trained across varying target speeds sampled uniformly in $[0, 0.5]$ m/s. These agents are given the target speed as observations. The evaluation procedure is the same as before, except we evaluate the same conditional policy for multiple target values. We make similar observations: a fixed penalty coefficient generally leads to higher speeds then the set target, and higher penalties. Interestingly, for higher target velocities, the actual velocity exceeds the target less, indicating that different values for $\alpha$ are required for different targets. As we learn multipliers that are conditioned on the target, we can track the target more closely, even for higher speeds. We also evaluate these models for a target speed outside out the training range. Performance degrades quite rapidly, with the constraint no longer satisfied, and at significantly higher

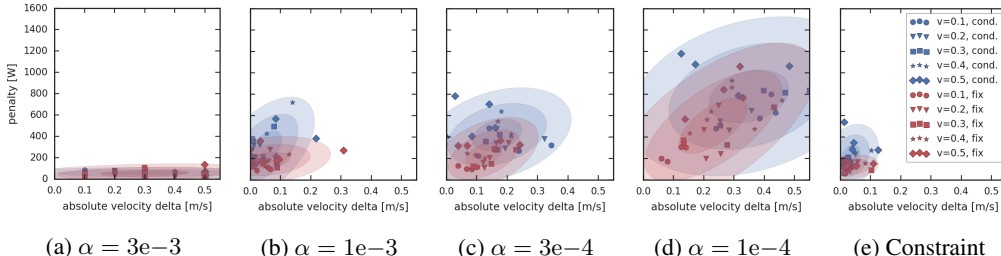

(a) $\alpha = 3\mathrm{e}{-3}$     (b) $\alpha = 1\mathrm{e}{-3}$     (c) $\alpha = 3\mathrm{e}{-4}$     (d) $\alpha = 1\mathrm{e}{-4}$     (e) Constraint

Figure 5: Comparison of the constrained optimization approach with baselines using a fixed penalty. Each data point shows the average absolute velocity delta and penalty for an agent optimized for a specific target velocity. The different ellipse shades show one to three standard deviations, both for the fixed (red) and the varying (blue) velocity setpoints. For each setting we train four agents. In the fixed target case, these are different models. In the conditional target case, these are evaluations of a single model conditioned on desired velocities.

cost. This can be explained by the way the policies change behavior to match the target speed. Generally the speed is changed by modulating the stride length. Increasing the stride length much further than observed during training, however, results in collisions occurring that were not present at lower speeds, and hence higher penalties. The same observation also explains why the penalties in the conditional case are higher than in the fixed case (final column in Table 3 vs. Table 2), as more distinct behaviors are needed to be optimal for each target velocity. This is likely a limitation of the relatively simple policy architecture, and improving diversity across goal velocities will be studied in future work.

Figure 5 extends the comparisons by plotting penalty over absolute velocity deltas for the different approaches. The plots show that finding a suitable weighting that works for all tasks and setpoints is difficult. While it is clear to identify values for $\alpha$ that are clearly too high or low, even for well-tuned values, performance over tasks can vary. Our approach as shown in Figure 5e is able to achieve a very consistent performance of low velocity tracking errors and low penalty across all tests. These results suggest, that our approach requires less problem specific tuning and is less sensitive to changes in the task. Therefore, a constraint-based approach can greatly reduce computationally expensive hyperparameter tuning. Videos showing some of the learned behaviors, both in the fixed and conditional constraint case, can be found at `https://sites.google.com/view/minitauriclr2019`.

## 5 CONCLUSION

In order to regularize behavior in continuous control RL tasks in a controllable way, we introduced a constraint-based approach that is able to automatically trade off rewards and penalties, and can be used in conjunction with any model-free, value-based RL algorithm. Specifically, we minimize the penalties with respect to a lower bound on the reward value. The constraints are applied in a point-wise fashion, for each state that the learned policy encounters, to allow for tighter control over the learned behavior. The resulting constrained optimization problem is solved using Lagrangian relaxation by iteratively adapting a set of Lagrangian multipliers, one per state, during training. By learning these state-dependent Lagrangian multipliers in the critic model alongside the value estimates of the policy, we can generalize multipliers to neighbouring states and efficiently and closely track the imposed bounds. The policy and critic can furthermore generalize across lower bounds by making the constraint value observable, resulting in a single bound-conditional RL agent that is able to dynamically trade off reward and costs in a controllable way. We applied our approach to a number of continuous control benchmarks and show that without some cost function, we observe high-amplitude and high-frequency control. Our method is able to reduce the control input significantly, sometimes without sacrificing average reward. In a simulated locomotion task with the Minitaur quadruped, we are able to minimize electrical power usage with respect to a lower bound on the forward velocity. We show that our method can achieve both lower velocity overshoot as well as lower power usage for different lower bounds compared to a baseline that uses a fixed coefficient for the penalty. We also learn a single, goal-conditioned policy that is able to move efficiently across a range of target velocities.

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

APPENDIX A: OPTIMIZATION DETAILS

**General algorithm** The general outline of the optimization procedure for Equation 5 is listed in Algorithm 1. The approach is compatible with any actor-critic algorithm; in the next paragraphs we detail the methods used in this paper for policy evaluation and optimization.

---

**Algorithm 1** Value constrained model-free control

---

1: **given** $Q_r\left(s, a; \psi_r^{(0)}, \phi^{(0)}\right)$, $Q_c\left(s, a; \psi_c^{(0)}, \phi^{(0)}\right)$, $\lambda\left(s; \psi_\lambda^{(0)}, \phi^{(0)}\right)$, $\pi(a|s; \theta^{(0)})$, with $\psi_i^{(0)}$,
   $\phi^{(0)}$ and $\theta^{(0)}$ initial weights, and replay buffer $\mathcal{D}$
2: **repeat**
3:     Execute $a \sim \pi(a|s; \theta^{(0)})$ and observe $s'$, $r(s, a)$, $c(s, a)$
4:     Add tuple $(s, a, s', r(s, a), c(s, a))$ to $\mathcal{D}$
5:     Sample batch $\mathcal{B}$ of tuples from $\mathcal{D}$
6:     **Critic update:**
7:     $L_r\left(\psi_r^{(k)}, \phi^{(k)}\right) = \mathbb{E}_\mathcal{B}\left[\text{valueLoss}\left(s, a, s', r(s, a), Q_r\left(s, a; \psi_r^{(k)}, \phi^{(k)}\right)\right)\right]$
8:     $L_c\left(\psi_c^{(k)}, \phi^{(k)}\right) = \mathbb{E}_\mathcal{B}\left[\text{valueLoss}\left(s, a, s', c(s, a), Q_c\left(s, a; \psi_c^{(k)}, \phi^{(k)}\right)\right)\right]$
9:     $L_\lambda\left(\psi_\lambda^{(k)}, \phi^{(k)}\right) = \mathbb{E}_\mathcal{B}\left[\max\left(0, \lambda\left(s; \psi_\lambda^{(k)}, \phi^{(k)}\right)\right)\left(\hat{Q}_r\left(s, a; \psi_r^{(k)}, \phi^{(k)}\right) - \bar{V}_r\right)\right]$
10:                                                                   $\triangleright$ Equation 5, no gradient through $\hat{Q}_r$
11:     $\psi_{r,c,\lambda}^{(k+1)}, \phi^{(k+1)} = \psi_{r,c,\lambda}^{(k)}, \phi^{(k)} - \eta_1 \cdot \nabla_{\psi_{r,c,\lambda}^{(k)}, \phi^{(k)}} \sum_{j \in \{r,c,\lambda\}} L_j\left(\psi_j^{(k)}, \phi^{(k)}\right)$
12:     **Policy update:**
13:     $\theta^{(k+1)} = \theta^{(k)} + \eta_2 \cdot \mathbb{E}_\mathcal{B}\left[\text{policyGradient}\left(\theta^{(k)}, s, a, Q_\lambda\left(s, a; \psi_{r,c,\lambda}^{(k)}, \phi^{(k)}\right)\right)\right]$
14: **until** stopping criterion is met
15: return $\psi_{r,c,\lambda}^{(k+1)}$, $\phi^{(k+1)}$ and $\theta^{(k+1)}$

---

**Policy Evaluation** Our method needs to have access to a Q-function for optimization. While any method for policy evaluation can be used, we rely on the Retrace algorithm (Munos et al., 2016). More concretely, we learn the Q-function for each cost term $Q_i(s, a; \psi_i, \phi)$, where $\psi_i, \phi$ denote the parameters of the function approximator, by minimizing the mean squared loss:

$$\min_{\psi_i, \phi} L(\psi_i, \phi) = \min_{\psi_i, \phi} \mathbb{E}_{\mu_b(s), b(a|s)}\left[\left(Q_i(s_t, a_t; \psi_i, \phi) - Q_t^{\text{ret}}\right)^2\right], \text{ with}$$

$$Q_t^{\text{ret}} = Q_i(s_t, a_t; \psi_i', \phi') + \sum_{j=t}^{\infty} \gamma^{j-t}\left(\prod_{k=t+1}^{j} c_k\right)\left[r_i(s_j, a_j) + \right.$$

$$\mathbb{E}_{\pi(a|s_{j+1})}[Q_i(s_{j+1}, a; \psi_i, \phi)] - Q_i(s_j, a_j; \psi_i', \phi')\bigg],$$

$$c_k = \min\left(1, \frac{\pi(a_k|s_k)}{b(a_k|s_k)}\right),$$

(7)

where $Q_i(s, a; \psi_i', \phi')$ denotes the output of a target Q-network, with parameters $\psi_i', \phi'$, that we copy from the current parameters after a fixed number of updates. Note that while the above description uses the definition of reward $r_i$ we learn the value for the costs analogously. We truncate the infinite sum after $N$ steps by bootstrapping with $Q_{\phi'}$. Additionally, $b(a|s)$ denotes the probabilities of an arbitrary behaviour policy, in our case given through data stored in a replay buffer.

We use the same critic model to predict all values as well as the Lagrangian multipliers $\lambda(s, \psi_\lambda, \phi)$. Following Equation 5, we hence also minimize the following loss:

$$\min_{\psi_\lambda, \phi} L(\psi_\lambda, \phi) = \mathbb{E}_{\mu_b(s)}\left[\min_{\lambda(s, \psi_\lambda, \phi) \geq 0} \mathbb{E}_{a \sim \pi}[Q_\lambda(s, a)]\right]$$

(8)

Our total critic loss to minimize is $\sum_i L\left(\boldsymbol{\psi}_i, \phi\right) + \beta \cdot L\left(\boldsymbol{\psi}_\lambda, \phi\right)$, where $\beta$ is used to balance the constraint and value prediction losses.

**Maximum a Posteriori Policy Optimization**     Given the Q-function, in each policy optimization step, MPO use expectation-maximization(EM) to optimize the policy. In the E-step MPO finds the solution to a following KL regularized RL objective; the KL regularization here helps avoiding premature convergence, we note, however, that our method would work with any other policy gradient algorithm for updating $\pi$. MPO performs policy optimization via an EM-style procedure. In the E-step a sample based optimal policy is found by minimizing:

$$\max_q \mathbb{E}_{\mu(s)}\left[\mathbb{E}_{q(a|s)}\left[Q_i\left(\boldsymbol{s}_t, \boldsymbol{a}_t; \boldsymbol{\psi}_i, \boldsymbol{\phi}\right)\right]\right]$$
$$s.t. \mathbb{E}_{\mu(s)}\left[\text{KL}(q(a|s), \pi_{old}(a|s))\right] < \epsilon. \tag{9}$$

Afterwards the parametric policy is fitted via weighted maximum likelihood learning (subject to staying close to the old policy) given via the objective:

$$\max_\pi \mathbb{E}_{\mu(s)}\left[\mathbb{E}_{q(a|s)}\left[\log \pi(a|s)\right]\right]$$
$$s.t. \ \mathbb{E}_{\mu(s)}\left[\text{KL}(\pi_{old}(a|s), \pi(a|s))\right] < \epsilon. \tag{10}$$

assuming a Gaussian policy (as in this paper) this objective can further be decoupled into mean and covariance parts for the policy (which in-turn allows for more fine-grained control over the policy change) yielding:

$$\max_\pi \mathbb{E}_{\mu(s)}\left[\mathbb{E}_{q(a|s)}\left[\log \pi(a|s)\right]\right]$$
$$s.t. \ C_\mu < \epsilon_\mu$$
$$C_\Sigma < \epsilon_\Sigma \tag{11}$$

$$\int \mu(s)\text{KL}(\pi_{old}(a|s), \pi(a|s)) = C_\mu + C_\Sigma, \tag{12}$$

where

$$C_\mu = \int \mu(s)\tfrac{1}{2}(\text{tr}(\Sigma^{-1}\Sigma_{old}) - n + \ln(\frac{\Sigma}{\Sigma_{old}}))ds,$$
$$C_\Sigma = \int \mu(s)\tfrac{1}{2}(\mu - \mu_{old})^T \Sigma^{-1}(\mu - \mu_{old})ds.$$

This decoupling of updating mean and covariance allows for setting different learning rate for mean and covariance matrix and controlling the contribution of the mean and co-variance to KL seperatly. For additional details regarding the rationale of this procedure we refer to the original paper Abdolmaleki et al. (2018).

**Hyperparameters**     The hyperparameters for the Q-learning and policy optimization procedure are listed in Table 4. We perform optimization of the above given objectives via gradient descent; using different learning rates for critic and policy learning. We use Adam for optimization.

Table 4: Overview of the hyperparameters used for the experiments.

| Parameter | Cart-pole | Humanoid | Minitaur |
|---|---|---|---|
| Hidden units policy | $100 - 100$ | $300 - 200$ | $300 - 200$ |
| Hidden units critic | $200 - 200$ | $400 - 300$ | $300 - 200$ |
| LSTM cells | - | - | 100 |
| Discount | 0.99 | 0.99 | 0.99 |
| Policy learning rate | $1e{-}5$ | $1e{-}5$ | $1e{-}5$ |
| Critic learning rate | $1e{-}4$ | $1e{-}4$ | $3e{-}4$ |
| Constraint loss scale ($\beta$) | $1e0$ | $1e0$ | $1e{-}3$ |
| Number of actors | 32 | 32 | 100 |
| E-step constraint($\epsilon$) | $1e{-}1$ | $1e{-}1$ | $1e{-}2$ |
| M-step constraint on $\mu$ ($\epsilon_\mu$) | $1e{-}2$ | $1e{-}2$ | $1e{-}4$ |
| M-step constraint on $\Sigma$ ($\epsilon_\Sigma$) | $1e{-}5$ | $1e{-}5$ | $1e{-}6$ |

## APPENDIX B: MINITAUR SIMULATION DETAILS

Table 5: Overview of the different model variations and noise models in the Minitaur domain. $\mathcal{N}(\mu, \sigma)$ is the normal distribution, $\mathrm{Lognormal}(\mu, \sigma)$ the corresponding log-normal. $\mathcal{U}(a, b)$ is the uniform distribution and $\mathcal{B}(p)$ the Bernouilli distribution.

| Parameter | Sample frequency | Description |
|---|---|---|
| Body mass | episode | global scale $\sim$ Lognormal $(0, 0.1)$, with scale for each separate body $\sim$ Lognormal $(0, 0.02)$ |
| Joint damping | episode | global scale $\sim$ Lognormal $(0, 0.1)$, with scale for each separate joint $\sim$ Lognormal $(0, 0.02)$ |
| Battery voltage | episode | global scale $\sim$ Lognormal $(0, 0.1)$, with scale for each separate motor $\sim$ Lognormal $(0, 0.02)$ |
| IMU position | episode | offset $\sim \mathcal{N}(0, 0.01)$, both cartesian and angular |
| Motor calibration | episode | offset $\sim \mathcal{N}(0, 0.02)$ |
| Gyro bias | episode | $\mathcal{N}(0, 0.001)$ |
| Accelerometer bias | episode | $\mathcal{N}(0, 0.01)$ |
| Terrain friction | episode | $\mathcal{U}(0.2, 0.8)$ |
| Gravity | episode | scale $\sim$ Lognormal $(0, 0.033)$ |
| Motor position noise | time step | $\mathcal{N}(0, 0.04)$, additional dropout $\sim \mathcal{B}(0.001)$ |
| Angular position noise | time step | $\mathcal{N}(0, 0.001)$ |
| Gyro noise | time step | $\mathcal{N}(0, 0.01)$ |
| Accelerometer noise | time step | $\mathcal{N}(0, 0.02)$ |
| Perturbations | time step | Per-step decay of 5%, with a chance $\sim \mathcal{B}(0.001)$ of adding a force $\sim \mathcal{N}(0, 10)$ in any planar direction |