# OpenReview forum: "Success at any cost: value constrained model-free continuous control"
_ICLR.cc/2019/Conference_

### Official Review · AnonReviewer1 · 2018-10-17

**Rating:** 6
**Confidence:** 4

**Review:**

This paper proposes an approach for mitigating issues associated with high-frequency/amplitude control signals that may be obtained when one applies reinforcement learning algorithms to continuous control tasks. The approach taken by the paper is to solve a constrained optimization problem, where the constraint imposes a (potentially state-dependent) lower bound on the reward. This is done by using a Lagrangian relaxation that learns the parameters of a control policy that satisfies the desired constraints (and also learns the Lagrange multipliers). The presented approach is demonstrated on a cart-pole swing-up task as well as a quadruped locomotion task.

Strengths:
+ The paper is generally clear and readable.
+ The simulation results for the Minitaur quadruped robot are performed using a realistic model of the robot.

Major concern:
- My biggest concern is that the technical contributions of the paper are not clear at all. The motivation for the work (avoiding high amplitude/frequency control inputs) is certainly now new; this has always been a concern of control theorists and roboticists (e.g., when considering minimum-time optimal control problems, or control schemes such as sliding mode control). The idea of using a constrained formulation is not novel either (constrained MDPs have been thoroughly studied since Altman (1999)). The technical approach of using a Lagrangian relaxation is the standard way one goes about handling constrained optimization problems, and thus I do not see any novelty there either. Overall, the paper does not make a compelling case for the novelty of the problem or approach.

Other concerns:
- For the cart-pole task, the paper states that the reward is modified "to exclude any cost objective". Results are then presented for this modified reward showing that it results in high-frequency control signals (and that the proposed constrained approach avoids this). I don't think this is really a fair comparison; I would have liked to have seen results for the unmodified reward function.
- The claim made in the first line of the abstract (applying RL algorithms to continuous control problems often leads to bang-bang control) is very broad and should be watered down. This is the case only when one considers a poorly-designed cost function that doesn't take into account realistic factors such as actuator limits.
- In the last paragraph of Section 3.3, the paper proposes making the lower-bound on the reward state-dependent. However, this can be tricky in practice since it requires having an estimate for Q_r(s,a) as a function of the state (in order to ensure that the state-dependent lower bound can indeed be satisfied).

Typos:
- Pg. 5, Section 3.4: "...this is would achieve..."
- Pg. 6: ...thedse value of 90..."

---

> ### Author Response · Authors · 2018-11-26
> **Author response**
>
> Thank you for your comments. Please find below our response to your questions and concerns.
>
> 1) Technical contributions
> We are glad that the reviewer agrees that we are tackling a long standing and important problem and acknowledge the fact that neither the definition of constrained MDPs nor the application of Lagrangian relaxation to solve these problems is novel by itself. We should have stated our exact technical contributions more clearly and have adapted the paper to do so. For completeness we will list these below:
> a) We introduce pointwise, per-state constraints to learn more consistent behavior compared a single global constraint, and regress the resulting state-dependent Lagrangian multipliers using a neural network to exploit generalization across similar states.
> b) Instead of recombining the reward and cost directly on the environment side and learning a single value estimate, we train a critic network to output both return and penalty value estimates as well as the Lagrangian multipliers themselves, effectively providing more structure to the critic. We only combine the different terms appropriately for the actor update.
> c) We show that we can train a single, bound-conditional policy that can optimize penalty across a range of bounds and can be used to dynamically trade off reward and penalty.
>
> 2) Comparison with the original benchmark reward
> We have extended the results on Cartpole to include the original reward as defined in the DM Control Suite (incl. bonus for low control). We found that compared to the original setting, our method is able to reduce the average control norm by over 50% across the entire episode, and by over 80% after the swingup phase, without significant reduction in the average return as measured without control bonus.
>
> 3) Claims about bang-bang control in continuous RL
> The reviewer is right in that the claim of RL often leading to bang-bang control is too strongly worded. This is only the case when the objective function is not well-designed and one is naively optimizing for success only. Designing a proper objective function is however often not trivial and more of an art, requiring several iterations to achieve the desired behavior. This work tries to remove some of the complexities in designing such a function.
>
> 4) State-dependent lower bound
> Defining a state-dependent bound is indeed not trivial and requires knowledge of what is feasible in the system, and as such we leave this up to future work. In this paper we have made the approximation that the state distribution is stationary and the discount is large enough to assume that the value is more or less constant. While this holds for locomotion tasks, this does not apply in e.g. the swingup phase of the cartpole task and as a result the penalty is completely ignored during this phase.

---

> > ### Comment · AnonReviewer1 · 2018-12-11
> > **Updated evaluation**
> >
> > I thank the authors for their comments and revisions. I have updated my evaluation to "6: marginally above acceptance threshold." The revision makes the paper's contributions clearer (which was my primary concern). I think that the paper makes contributions that are potentially interesting to researchers in reinforcement learning (but I still don't think that the contributions are exceptionally strong). I am still concerned about the issues I had raised before w.r.t. state-dependent lower bounds and I still think that many of the issues the paper tackles (i.e., mitigating bang-bang control) are often relatively easy to tackle with heuristic methods (like changing the cost function) -- this is what roboticists tend to do for hardware implementations of optimal controllers. These are the reasons for my evaluation.

---

### Official Review · AnonReviewer2 · 2018-11-02
**value constrained model-free continuous control**

**Rating:** 5
**Confidence:** 4

**Review:**

This paper uses constrained Markov decision processes to solve a multi-objective problem that aims to find the correct trade-off between cost and return in continuous control. The main technique is Lagrangian relaxation and experiments are focus on cart-pole and locomotion task.

Comments:

1) How to solve the constrained problem (8) is unclear. It is prefer to provide detailed description or pseudocode for this step.

2) In equation (8), lambda is a trade-off between cost and return. Optimization on lambda reduces burdensome hyperparameter selection, but a new hyperparameter beta is introduced. How do we choose a proper beta, and will the algorithm be sensitive to beta?

3) The paper only conducts comparison experiments with fixed-alpha baselines. The topic is similar to safe reinforcement learning. Including the comparison with safe reinforcement learning algorithms is more convincing.

---

> ### Author Response · Authors · 2018-11-26
> **Author response**
>
> Thank you for your comments. Please find below our response to your questions and concerns.
>
> 1) Pseudocode
> We apologise that the optimization procedure was unclear. We have added pseudocode of the general optimization procedure in Appendix A.
>
> 2) Hyperparameter selection
> The reviewer is completely right that we are removing one hyperparameter by introducing another. However, there are two reasons why this might still be beneficial: one is that the penalty coefficient is now effectively dynamic and can change during training, ensuring higher chances of finding a good solution. Second, by elevating the hyperparameter one level up, we hope that the learning is indeed less sensitive to its specific setting. Indeed, we found in practice that we get similar results for \beta within some orders of magnitude, which requires significantly less tuning compared to a fixed \alpha.
>
> 3) Relation to safe reinforcement learning
> It is indeed the case that constrained MDPs are often considered in safe RL. In those cases there is generally an upper bound on a penalty function that should never be exceeded, including during training itself. These algorithms generally restrict policy updates to remain within the constraint-satisfying regime. While our approach can similarly be applied to upper bounds on penalties, there’s unfortunately no guarantee that the constraints will be satisfied at every moment during training, but only at convergence. As such it is not clear how these methods would apply to our specific experimental setups.

---

> > ### Comment · AnonReviewer2 · 2018-12-12
> > **response**
> >
> > I have read the author response and my opinion remains the same.
> >
> > 2) Hyperparameter selection
> >
> > The authors did not remove the hyperparameter completely, but introduced a new hyperparameter and stated that hyperparameter could make the learning more robust. The robustness of the new hyperparameter has not been verified.
> >
> > 3) Relation to safe reinforcement learning
> >
> > It did some modification on the CMDP and stated the proposed approach could satisfy the desired constraints in the abstract. But in the rebuttal, the authors stated that "there’s unfortunately no guarantee that the constraints will be satisfied at every moment during training". It is confused whether the proposed method could guarantee the desired constraints and how to obtain "a constant speed is more desirable than a fluctuating one" as stated in section 3.3.

---

### Official Review · AnonReviewer4 · 2018-11-07

**Rating:** 7
**Confidence:** 4

**Review:**

This paper proposes a model free reinforcement learning algorithm with constraint on reward, with demonstration on cartpole and quadruped locomotion.

strength: (1) challenging examples like the quadruped.
                (2) result seems to indicate the method is effective

There are several things I would like the authors to clarify:
(1) In section 3.2, why is solving (4) would give a "exactly the desired trade-off between reward and cost"? First of all, how is the desired trade-off defined? And how is (4) solved exactly? If it is solved iteratively, i.e, alternating between the inner min and outer max, then during the inner loop, wouldn't the optimal value for \lambda be infinity when constrained is violated (which will be the case at the beginning)? And when the constrained is satisfied, wouldn't \lambda = 0? How do you make sure the constrained will still be satisfied during the outer loop since it will not incurred penalty(\lambda=0). Even if you have a lower bound on \lambda, this is introducing additional hyperparameter, while the purpose of the paper is to eliminate hyperparamter?
(2) In section 3.2, equation 6. This is clearly not a convex combination of Qr-Vr and Qc, since convex combination requires nonnegative coefficients. The subtitle is scale invariance, and I cannot find what is the invariance here (in fact, the word invariance\invariant only appears once in the paper). By changing the parametrization, you are no longer solving the original problem (equation 4), since in equation (4), the only thing that is related to \lambda is (Qr-Vr), and in (6), you introduce \lambda to Qc as well. How is this change justified?
(3)If I am not mistaken, the constrained can still be violated with your method. While from the result it seems your method outperforms manually selecting weights to do trade off,  I don't get an insight on why this automatic way to do tradeoff is better. And this goes back to "exactly the desired trade-off between reward and cost" in point(1), how is this defined?
(3) The comparison in the cartpole experiment doesn't seem fair at all, since the baseline controller is not optimized for energy, there is no reason why it would be comparable to one that is optimized for energy. And why would a controller " switch between maximum and minimum actuation is indeed the optimal solution" after swingup? Maybe it is "a" optimal solution, but wouldn't a controller that does nothing is more optimal(assuming there is no disturbance)?
(4)For Table I, the error column is misleading. If I understand correctly, exceeding the lower bound is not an error (If I am wrong, please clarify it in the paper). And it is interesting that for target=0.3, the energy consumption is actually the lowest.
(5)Another simple way to impose constrained would be to terminate the episode and give large penalty, it will be interesting to see such comparison.

minor points:
* is usually used for optimal value, but is used in the paper as a bound.

---

> ### Author Response · Authors · 2018-11-26
> **Author response**
>
> Thank you for your comments. Please find below our response to your questions and concerns.
>
> 1) Definition of desired trade-off and practical optimization
> Using “trade-off” in Section 3.2 is indeed an unfortunate choice of words. What we intended to explain was that when the gradient with respect to \lambda is zero, either \lambda itself is zero (and optimizing for the penalty does not worsen the return), or the average value is exactly the bound and we satisfy the constraint, while still minimizing the penalty.
> More broadly, this work situates itself in multi-objective optimization,where the objectives are counteracting at least part of the time, meaning that in order to gain in one objective, one has to lose in the other. It is here that this “trade-off” comes into play. Different ratios of the objective will (generally) lead to different results. It is however a priori not trivial to define the right ratio for the desired behavior (e.g. certain minimum speed, or maximum power usage). Formulating the problem in terms of constraints on either objective is often more intuitive.
> As noted correctly by the reviewer, a full optimization of \lambda in Equation (4) in the inner loop would either lead to \lambda being zero or infinity. In practice, however, one generally updates \lambda only incrementally before optimizing the policy w.r.t the updated \lambda. Ideally, one would optimize the policy until convergence before updating \lambda, as one can effectively switch the inner and outer optimization step, however we found that in practice this is not necessary and instead perform one update to \lambda for each policy update.
> We have added pseudocode for the exact optimization procedure in appendix to make this more clear.
>
> 2) Convex combination
> If we define L_1 = Q_r-V_r* and L_2 = -Q_c, then L_1 and L_2 are two objectives that we are trying to maximize simultaneously, but in different ratios depending on the value of \lambda’. By changing base to L_1 and L_2 we do effectively get a convex combination of L_1 and L_2. As such, Q_\lambda’ is always smaller or equal to max(L_1, L_2). If we don’t add this normalization step Q_\lambda can become much larger, with increasing \lambda, than either L_1 or L_2, which we have found can lead to stability problems in the policy update. Another way to look at this is that in order to optimize the return, we want to be able to suppress the penalty until we meet the lower bound.
> It is indeed correct that the optimization objective does change when optimizing Equation (6) as is. What should do instead is only consider the gradient w.r.t. \lambda’ coming from the first term in the numerator. The way we implemented this in practice ensured this implicitly, and it was an oversight of us not to mention this in the original submission, our apologies.
>
> 3) Benefit of automatic trade off
> It is indeed the case that during training the constraint can still be violated. Moreover, in the way we formulated it with a lower bound on the return, this will most definitely be the case. It is only at convergence, when the gradient w.r.t. \lambda is 0, that the constraint is strictly satisfied.
> The main benefit of doing this trade-off automatically is that one can specify the desired behavior in terms of a value in one of the objectives, instead of trying out different ratios and verifying the result. Moreover, there is the added flexibility of the ratios changing during training itself, which may help to overcome issues with exploration when the penalty dominates the reward too much at the start of learning.

---

> > ### Comment · AnonReviewer4 · 2018-11-26
> > **Thanks for the clarification**
> >
> > Thanks for the clarification.
> > Further comment:
> > (2) So the convex combination is between (Qr-Vr) and -Qc, not (Qr-Vr) and Qc (this is what is in the paper), please update the paper accordingly. I still don't get what is the invariance is here.
> >
> > Other than that, i think my concerns are addressed properly with the current version of the paper and believe it will be of interest to many reinforcement learning researchers and recommend acceptance.

---

> > > ### Author Response · Authors · 2018-11-26
> > > **Corrected error**
> > >
> > > Thank you for updating your review and pointing out this error in the paper, we should have spotted this sooner! We have uploaded a corrected version.

---

> ### Author Response · Authors · 2018-11-26
> **Author response**
>
> 4) Comparing the constrained with unconstrained optimization and the optimality of bang-bang control
> We agree that comparing the resulting control signal of the constrained with the unconstrained case as is, is unfair. The goal here was solely to illustrate the issue that unless some form of penalty is included in the optimization perspective, agents often learn bang-bang control. In this work, we want to automatically tune the magnitude of this penalty based on some bound on the success rate. Specifically for cartpole, we show that we can greatly reduce the average control norm without sacrificing task performance. We have added additional results to reflect this, and have also added a comparison with the reward as originally defined in the DM Control Suite.
> Regarding the optimality of bang-bang control, we meant to refer to the swingup phase itself, not after, apologies for the inclarity. In order to swing up the cartpole as quickly as possible, applying maximum control is indeed the best thing to do. This is related to minimum-time optimal control, where, based on Pontryagin's maximum principle, the optimal control value to reach a certain state in the minimum amount of time will always be an extreme value within the admissible range of controls. As to why we still observe bang-bang control after the swingup phase, this is not clearly understood. Perhaps the minimum-time optimal control principle still holds here, as the policy is generally never able to exactly match the perfect balancing state. Other plausible reasons are that this is a result of exploration noise. As the reviewer notes, bang-bang control is only a solution out of a possibly large set. We however find in practice that without any additional objective, more often than not policies learn this style of control.
>
> 5) Velocity error
> The reviewer is correct in that the term “error” is badly chosen in this case, as it is indeed not necessarily required to stick to close to the return bound in order to optimize the penalty. A more appropriate term is “overshoot”, and we have adapted the paper to use this wording. In the context of this experiment however, the reward and cost are strictly antagonistic, so the smaller the overshoot with the bound the better.
>
> 6) Episode termination on large penalty
> Ending the episode when the constraint is violated would indeed be an alternative to solving the constrained optimization problem in some cases. There are two conditions however: the constraint has to be put on the penalty and has to be satisfied at the start of learning (or each episode will terminate instantly), and the agent does not have to learn to recover from situation where it can not satisfy the constraint. For example, in the case of an extreme disturbance, the agent might have to output more power than the constraint would allow.

---

### Comment · Area_Chair1 · 2018-11-20
**authors -- chance to respond?**

Thanks for all the reviews.
If the authors wish to respond, this would be a good time.
-- area chair

---

### Author Response · Authors · 2018-11-26
**Author response**

We would first of all like to thank the reviewers for their insightful comments and greatly appreciate the feedback. Our apologies for the unclarities raised and we will try to answer all questions as clear as possible in response to each individual review.

Based on the reviewers’ comments, the main changes to the paper are the following:
1) We explicitly report average reward and control penalty for the cart-pole swingup task, and compare the constrained and unconstrained cases with a policy trained with the original task reward that included a fixed control penalty. We show that we can achieve the same return for a significantly less penalty.
2) We have added additional results on two other, more challenging, continuous control benchmark tasks: humanoid stand and walk. For the stand task, we observe a similar trend as for cart-pole, where the constraint-based approach is able to achieve a much lower penalty without sacrificing task performance. For the walk task, we satisfy the imposed lower bound.
3) Stated the technical contributions more explicitly in Section 1.
4) Added pseudocode for the optimization procedure in Appendix A.

---

### Comment · Area_Chair1 · 2018-12-08
**further input needed from R1 & R2**

The discussion period is ending soon (Dec 9?).
The authors have replied in some detail.
Feedback from R1 & R2 on the replies would be particularly useful.
Do include a short summary of pros & cons from your perspective if you can.

Your time is very much appreciated.
-- area chair

---

### Meta-Review · Area_Chair1 · 2018-12-14
**some novelty;  muted endorsements; solid writing and results; revisit?**

**Confidence:** 3
**Recommendation:** Reject

**Metareview:**

Strengths:  The paper introduces a novel constrained-optimization method for RL problems.
A lower-bound constraint can be imposed on the return (cumulative reward),
while optimizing one or more other costs, such as control effort.
The method learns multiple
The paper is clearly written.  Results are shown on the cart-and-pole, a humanoid, and a realistic Minitaur
quadruped model.  AC: Being able to learn conditional constraints is an interesting direction.

Weaknesses:  There are often simpler ways to solve the problem of high-amplitude, high-frequency
controls in the setting of robotics.
The paper removes one hyperparameter (lambda) but then introduces another (beta), although beta
is likely easier to tune. The ideas have some strong connections to existing work in
safe reinforcement learning.
AC: Video results for the humanoid and cart-and-pole examples would have been useful to see.

Summary:   The paper makes progress on ideas that are fairly involved to explore and use
(perhaps limiting their use in the short term), but that have potential,
i.e., learning state-dependent Lagrange multipliers for constrained RL. The paper is perfectly fine
technically, and does break some new ground in putting a particular set of pieces together.
As articulated by two of the reviewers, from a pragmatic perspective, the results are not
yet entirely compelling. I do believe that a better understanding of working with constrained RL,
in ways that are somewhat different than those used in Safe RL work.

Given the remaining muted enthusiasm of two of the reviewers, and in the absence of further
calibration, the AC leans marginally towards a reject. Current scores: 5,6,7.
Again, the paper does have novelty, although it's a pretty intricate setup.
The AC would be happy to revisit upon global recalibration.